# Predicting Postoperative Lung Cancer Recurrence and Survival Using Cox Proportional Hazards Regression and Machine Learning

**DOI:** 10.3390/cancers17010033

**Published:** 2024-12-26

**Authors:** Lucy Pu, Rajeev Dhupar, Xin Meng

**Affiliations:** 1Department of Bioengineering, University of Pennsylvania, Philadelphia, PA 19104, USA; lpu@seas.upenn.edu; 2Department of Cardiothoracic Surgery, Wake Forest University, Winston-Salem, NC 27109, USA; rdhupar@wakehealth.edu; 3Department of Radiology, School of Medicine, University of Pittsburgh, Pittsburgh, PA 15213, USA

**Keywords:** lung cancer recurrence, CT biomarkers, risk prediction, Cox regression, machine learning

## Abstract

Surgical resection is the optimal treatment for early-stage lung cancer; however, recurrence rates remain high. Predicting recurrence accurately remains a significant challenge. This study aims to identify image biomarkers from preoperative chest CT scans to predict recurrence in patients undergoing lung resection. Using data from 309 patients, the study evaluates Cox proportional hazards regression analysis alongside machine learning (ML) methods for recurrence prediction. Key risk factors identified include surgical procedure, TNM staging, lymph node involvement, body composition, and tumor characteristics. Both Cox and ML models demonstrated comparable performance, achieving AUCs of 0.75–0.77, highlighting the potential of CT-based biomarkers for predicting recurrence.

## 1. Introduction

Lung cancer remains the leading cause of cancer-related mortality in the United States (US), accounting for 22% of all cancer deaths. It claims more lives than the combined total of the next three cancers: colon, breast, and prostate cancer [1]. Non-small-cell lung cancer (NSCLC) represents approximately 85% of all lung cancer cases [2]. For patients with stage I-IIIA NSCLC, surgical resection is widely regarded as the optimal curative option [3], constituting over 26% of all lung cancer therapies [4,5,6,7,8]. The increasing adoption of low-dose computed tomography (LDCT) for lung cancer screening is expected to detect more early-stage lung cancers, expanding the pool of patients eligible for curative-intent treatment like surgical resection.

Unfortunately, the recurrence rate for lung cancer is unacceptably high even after optimal surgical treatment [9]. Studies have reported a 30–55% rate of recurrence after surgery for early-stage NSCLC [10,11]. This high recurrence rate not only causes anxiety among lung cancer survivors but also significantly impacts their quality of life and the well-being of their families. Furthermore, postoperative recurrence is often distant (i.e., metastasis) [11,12] and is associated with poor prognosis and high mortality [11,13,14]. Survival rates after lung cancer recurrence have been reported as (1) one-year survival rate of 28% [15], (2) two-year survival rate of 11% [15], (3) three-year survival rate of 13% [16], and (4) five-year survival rate of 13% [17]. Therefore, understanding the underlying mechanism of lung cancer recurrence is of great importance. This knowledge would allow for individualized surveillance strategies, personalized treatment plans, and enhanced shared decision-making, ultimately improving the outcome of lung cancer patients.

A variety of factors associated with lung cancer recurrence have been explored, including (1) patient- and tumor-specific characteristics, such as smoking history and status, age, and cancer stage [18,19], (2) the role of systemic inflammation [20,21], (3) the influence from perioperative elements like transfusion, hypothermia, surgical trauma, and anesthesia [22,23], (4) molecular-level biomarkers [24,25], and (5) radiomic biomarkers [26]. While much attention has been given to the first four areas, radiomic studies have largely focused on postoperative histopathology images [26,27]. Despite substantial efforts, limited progress has been made in reliably predicting which patients are at risk of postoperative lung cancer recurrence. It is evident that postoperative recurrence is caused by a complex interplay of multiple conditions and mechanisms, but it is difficult to enumerate all the potential factors contributing to recurrence. An approach that can concisely and accurately identify and integrate the critical factors related to recurrence to enable easy, early, and accurate prediction of lung cancer recurrence would be a significant advancement in clinical practice.

We believe that cancer and its recurrence are caused by the long-term accumulation of genetic, epigenetic, and environmental factors. Internal factors related to recurrence can be conceptualized into three levels, including (1) body: the body provides the global environment for cancer development and metastases; (2) tumor-body interaction: tumor cells must overcome various biological barriers to circulate throughout the body and establish metastasis in new locations; and (3) tumor: tumor-specific characteristics influence its aggressiveness and postoperative behavior. An individual’s body composition is, in part, a reflection of the individual’s long-term habits and lifestyle choices, such as physical activity, exercise, and diet. It is not surprising that numerous chronic conditions, such as cardiovascular disease [28], diabetes [29], cancers [30], and osteoporosis [31], are closely associated with body composition. Studies have consistently demonstrated the importance of body composition in both long- and short-term health outcomes [32,33,34,35]. However, to our knowledge, limited effort has been dedicated to investigating the association between body composition and postoperative lung cancer recurrence. Traditional methods for measuring body composition include body mass index (BMI), waist circumference (WC), bioimpedance, and dual-energy x-ray absorptiometry (DXA). Compared to these traditional approaches, volumetric imaging modalities (e.g., CT scan) provide an opportunity to compute a more extensive, multi-tissue assessment of body composition [36,37].

Based on the above consideration, a multi-level radiomics approach is proposed with a special emphasis on body composition to systematically capture factors associated with postoperative lung cancer recurrence. This approach uses chest CT scans to quantify radiomic features corresponding to (1) body composition, (2) lung characteristics, and (3) tumor characteristics. The radiomic features are then integrated with patient demographics, surgical approach, and pathological data to create a model for predicting postoperative recurrence and survival in lung cancer patients.

## 2. Methods and Materials

### 2.1. Study Cohort

This study utilized a cohort of 309 lung cancer patients from the Lung Cancer Database, managed by the Department of Cardiothoracic Surgery at the University of Pittsburgh (Table 1). The inclusion criteria were (1) curative-intent lung resection, (2) pathologically confirmed NSCLC, (3) preoperative chest CT scans with an image thickness of 2.5 mm or less, (4) preoperative PET-CT scans from the neck to the thigh, and (5) a minimum 5 years of follow-up. The pre-treatment chest CT and PET-CT scans closest to the surgery date were used to quantify tumor characteristics and body composition, respectively. Clinical data, including age, sex, smoking history, surgical approach, and pathological stage, were collected through a review medical record. All data were de-identified and re-identified with a unique study ID number by an honest broker to ensure privacy. This study was approved by the University of Pittsburgh Institutional Review Board (IRB) (IRB#: 20100305).

Chest CT and PET-CT scans used in this study were acquired over a period of more than ten years, utilizing a range of imaging protocols and scanners. The CT scanners employed included those manufactured by GE HealthCare with models of LightSpeed-VCT, Optima-CT660, and LightSpeed-Ultra (Waukesha, WI, USA) and by Siemens Healthineers with models of Emotion and Emotion-Duo (Erlangen, Germany). All chest CT scans were performed without the use of contrast agents, with patients positioned supine and instructed to hold their breath at full inspiration. The images were reconstructed to capture the full lung field using a 512 × 512 pixel matrix, with multiple reconstruction algorithms applied. Pixel dimensions in the in-plane direction ranged from 0.55 mm to 0.82 mm, and the image thickness varied from 0.625 mm to 2.5 mm. For PET-CT imaging, systems such as Discovery ST, Emotion, and Emotion-DUO were used, typically covering regions from the head to the thigh. The CT scans within the PET-CT exams had a 512 × 512 matrix, while the PET scan images had a 128 × 128 matrix. The in-plane resolution of the CT portions from the PET-CT exams ranged from 4.1 mm to 5.5 mm, with image thickness varying from 1.25 mm to 3.75 mm.

### 2.2. Algorithm Overview

The multi-level radiomics strategy has three key stages (Figure 1). The initial stage involves the calculation of radiomic features from CT images at three levels: (1) global—body level, which encompasses overall body characteristics; (2) regional—lung level, which pertains to characteristics specific to the lungs; and (3) local—tumor level, which relates to unique characteristics of the tumor.

The second stage uses a univariate analysis to identify variables significantly associated with postoperative recurrence and survival. The third stage is the development of multivariate prediction models based on Cox hazard analysis and machine learning approaches. These models integrate radiomic features with patient demographics and clinicopathological information.

### 2.3. Multi-Level Radiomic Features

#### 2.3.1. Body Composition

We have developed a convolutional neural network (CNN)-based solution [38] that can automatically segment five different body composition tissues depicted on CT images, including visceral adipose tissue (VAT), subcutaneous adipose tissue (SAT), intermuscular adipose tissue (IMAT), skeletal muscle (SM), and bone (Figure 2). One unique characteristic of this fully 3D algorithm is its generic ability to automatically segment five different body tissues on any CT images regardless of the body region from neck to thigh. The generic characteristic of our solution is primarily attributed to the diverse dataset utilized for machine learning, which encompasses CT scans obtained through a variety of protocols and covers the whole body, chest, and abdominal regions. In contrast, the majority of existing algorithms can only segment one to three types of body tissues on a single or few cross-sectional image slices of CT scans. A relatively comprehensive comparison with other available methods can be found in our publication [38].

In this study, the chest CT scans were processed using this fully 3D algorithm, and the body tissues were quantified using volume, mean density, and mass. The total volume and total mass of all five body tissues were calculated, as well as their piecewise ratios, such as the ratio of SAT to total fat and the ratio of fat to non-fat.

#### 2.3.2. Lung Characteristics

The lungs were segmented on chest CT scans, and the lung volume and density (Hounsfield unit (HU)) were computed using in-house software [39,40,41,42,43,44] (Figure 3). Regions associated with emphysema were quantified using a threshold of −950 HU. The extent of emphysema was calculated as the percentage of emphysema volume relative to the total lung volume. The pulmonary arteries and veins were segmented, and their volume ratios were calculated.

#### 2.3.3. Tumor Characteristics

Lung tumors depicted on the chest CT scans were automatically segmented using in-house software (Figure 4). The software quantified ten CT image features: volume, mean density, surface area, maximum diameter, mean diameter, solidness, mean diameter of the solid component, cavity ratio, calcification volume, and irregularity. The solid component of a nodule was determined using a threshold of −300 HU, and the cavity component was determined using a threshold of −910 HU. The irregularity of a nodule was calculated as the ratio of its surface area to its volume. The calcification volume was computed as the volume of the nodule with a density greater than 200 HU.

All the quantitative image analyses described above were conducted using our in-house software, which operates on Windows systems. The underlying algorithms have been previously published [38,39,40,41,42,45,46].

### 2.4. Univariate Cox Proportional Hazards Analysis

A univariate Cox proportional hazards (PHs) regression analysis [47] was conducted to examine the relationship between various factors and the risk of recurrence and survival following lung cancer surgery. Both recurrence-free survival (RFS) and overall survival (OS) were analyzed. RFS is defined as the time from surgery to the first recurrence, regardless of its location. RFS was further categorized into local/regional and distance RFSs. Local/regional RFS is the time from surgery to the first local or regional recurrence, while distant RFS refers to the time from surgery to the first distant recurrence. OS is defined as the interval from surgery to death or to the point of data collection for living patients. Additionally, distant OS refers to the OS of patients with distant recurrence, while local/regional OS is the OS of patients with local or regional recurrence. These factors included body composition, lung and tumor characteristics, clinicodemographic factors, and histopathologic information. To address the skewness in the volume-related parameters, log-transformation was applied. On the other hand, the density parameters were nearly normally distributed, so they were only rescaled for easier interpretation. The hazard ratios (HRs) for each normalized variable were calculated to assess the individual significance of each factor in relation to patient outcomes. Statistical significance was defined as a *p*-value of less than 0.05.

### 2.5. Postoperative Recurrence and Survival Prediction Modeling

Two distinct approaches were employed to discriminate patients who experience postoperative lung cancer recurrence and those who do not. A collinearity assessment was carried out prior to the multivariate Cox regression to identify and eliminate any highly correlated variables found to be significant in the univariate analysis. Only independent variables that were deemed to be significant in the univariate analysis were integrated into the models. The multivariate Cox proportional hazards (PHs) regression model is a measure of the relative risk of an event occurring at a certain time given certain covariates or risk factors. The cohort only has right-censored data; namely, some observations are only known to have occurred after a certain time point. The machine learning methods, logistic regression, support vector machine (SVM) [48], and random forest [49], were used to integrate the various factors obtained from multivariate Cox regression analysis and predict whether recurrence or survival will occur before or after a specific time point (e.g., 2 or 5 years). The output of these methods was the probability of recurrence or survival from the date of surgical resection.

### 2.6. Performance Validation

The performance of the prediction models was assessed by evaluating their ability to accurately discriminate patients who would experience recurrence and those who would survive up to 2 and 5 years following surgical resection. This was achieved using the 5-fold cross-valuation method, which ensures robust model evaluation by dividing the dataset into five subsets. Each subset is used as a test set once, while the remaining subsets are used for training, ensuring all data contribute to both training and testing. To quantify model performance, the area under a receiver operating characteristic (ROC) curve (AUC) was used as the primary performance metric. AUC is a widely accepted indicator of a model’s discriminatory ability, where higher values reflect better performance. Additionally, 95% confidence intervals (CIs) were computed for AUC to provide an estimation of the precision and reliability of the performance measure. To assess whether there was a significant statistical difference between the performance of the two prediction models, DeLong et al.’s method [50] was used to compare two ROC curves. The MedCalc software (v20.015) was used to perform these statistical analyses. In all analyses, statistical significance was set at a *p*-value of less than 0.05.

## 3. Results

### 3.1. Overall Postoperative Recurrence-Free Survival (RFS) and Overall Survival (OS) Analyses

Postoperative RFS and OS were relatively low in this study. Among the 309 patients with non-small-cell lung cancer (NSCLC), 70.9% (219/309) experienced cancer recurrence after surgery. Of these, 20.1% (44/219) had local or regional recurrence, while 79.9% (175/219) experienced distant recurrence. Notably, 88.9% (80/90) of the patients who did not experience recurrence ultimately passed away (Table 1). Kaplan–Meier analysis indicated a 5-year RFS rate of 31.6% (Figure 5a). Patients who experienced local or regional recurrence post-surgery had significantly better 5-year RFS and OS compared to those with distant recurrence (Figure 5b,d). Furthermore, significant differences in RFS and OS were observed based on the location of recurrence. Patients who had recurrence within the thoracic region exhibited notably longer RFS and OS compared to those with recurrence in other organs (Figure 5c,e).

### 3.2. Univariate Analyses of Local, Regional, and Distant RFS

In the univariate analysis, several variables, including demographic factors, surgical procedure, cancer stage, nodal involvement, body composition, vascular characteristics, and tumor traits, were significantly associated with RFS. Older patients exhibited significantly longer RFS times compared to younger individuals. Those who underwent lobectomy had notably better RFS outcomes compared to patients who received other surgical interventions. Additionally, current or former smokers experienced significantly shorter distant RFS, while higher BMI values were linked to shorter local/regional RFS (Table 2).

Patients with higher cancer stages or lymph node involvement exhibited significantly shorter RFS compared to those with lower stages or no lymph node involvement. Histological subtypes, particularly adenocarcinoma and squamous cell carcinoma, were significantly associated with longer RFS (Table 3).

In terms of body composition, higher values for SAT density, IMAT density, and SM density were all significantly associated with shorter RFS. Interestingly, increased pulmonary artery volumes, vein volumes, tumor volume, pleural area, and tumor irregularity were significantly associated with shorter RFS times. Lower values for tumor ground glass opacity were also significantly associated with shorter RFS times. In patients with local/regional recurrence, BMI, surgical procedure, and lymph node involvement, histological subtype, and SAT volume were significantly associated with RFS time (Table 2, Table 3 and Table 4). For patients with distant recurrence, smoking status, surgical procedure, cancer stage, lymph node involvement, histological subtype, body composition, and tumor characteristics were significantly associated with RFS times (Table 2, Table 3, Table 4 and Table 5). Also, interestingly, higher SAT volumes were linked to shorter local/regional RFS, while patients with higher SAT volumes had a significantly longer distant RFS (as seen in Table 4).

In Table 2, Table 3, Table 4 and Table 5, the significant variables were bolded. Variables associated with longer RFS have a hazard ratio less than 1, while those associated with shorter RFS have a HR greater than 1.

Pulmonary artery volume, pulmonary vein volume, tumor volume, and tumor irregularity were significantly associated with shorter OS times (Table 9). Lower values for decreased tumor ground glass opacity were significantly associated with shorter OS times. In patients with distant recurrence, BMI, surgical procedure, lung cancer stage, lymph node involvement, histological subtype, body composition, and tumor characteristics were significantly associated with OS times (Table 6, Table 7, Table 8 and Table 9). In Table 6, Table 7, Table 8 and Table 9, the significant variables were bolded. Variables associated with longer OS have a hazard ratio less than 1, while those associated with shorter OS have a HR greater than 1.

### 3.3. Multivariate RFS Analyses and Machine Learning Prediction Modeling

In the multivariate analyses of RFS, several factors, including surgical procedure, cancer stage, lymph node involvement, body composition, and vascular variables, were significantly associated with RFS (Table 10). Specifically, patients who underwent lobectomy exhibited significantly longer RFS time compared to those who underwent other types of surgical procedures. Variables significantly associated with shorter RFS include higher cancer stage, lymph node involvement, higher value of SM density, and higher pulmonary vein volume. Among patients with local or regional recurrence, the type of surgical procedure was significantly associated with RFS (Table 10). For patients with distant recurrence, surgical procedures, cancer stages, body composition, and vascular metrics were significantly associated with RFS (Table 10).

In the multivariate analyses of OS, lymph node involvement, body composition, and pulmonary artery variables were significantly associated with OS times (Table 11). Variables significantly associated with shorter OS were higher cancer stage, lymph node involvement, higher value of SM density, and higher pulmonary vein volume. Among patients with distant recurrence, OS was significantly influenced by cancer stage, body composition, and vascular metrics (Table 11).

The performance of the Cox PH model was evaluated for predicting 2- and 5-year postoperative recurrences and compared to three machine learning models, including support vector machine (SVM), random forest (RF), and logistic regression (LR). All these models were able to significantly differentiate patients who did or did not experience lung cancer recurrence within 2- or 5-years post-surgery (Figure 6). The random forest method demonstrated the highest performance in the 2-year recurrence analysis (AUC = 0.737, 95% CI: 0.680–0.794), although its performance was not significantly better than the other models. Similarly, the Cox PH model achieved the best performance in the 5-year recurrence analysis (AUC = 0.770, 95% CI: 0.714–0.826), but it was also not significantly superior to the other models. Overall, no significant differences were observed among the four modeling approaches (*p* > 0.05).

## 4. Discussion

A thorough investigation was carried out to identify the factors associated with postoperative recurrence and survival in lung cancer patients. The analysis focused on a range of variables, including patient demographics, surgical procedure, stage of lung cancer, involvement of lymph nodes, body composition, vascular metrics, and tumor characteristics. Several of these factors were found to be significantly associated with RFS and overall OS. The variables that demonstrated significance in the univariate analysis were integrated into predictive models to predict recurrence using a few different approaches (e.g., Cox regression and machine learning). To our knowledge, no prior studies have explored the detailed relationship between body composition, pulmonary vasculature, and lung cancer recurrence in the manner presented in this study. The models designed to identify patients who did or did not experience postoperative recurrence included variables that have not been previously investigated or objectively assessed using a computer algorithm. A distinctive feature of this study is its reliance on chest CT scans which are routinely obtained as part of standard care for lung cancer patients. This eliminates the need for additional radiation exposure or costly procedures, offering a practice and accessible method for recurrence prediction. Furthermore, postoperative recurrence is a challenge not only in lung cancer but also in other cancers, such as liver, colon, and esophageal cancer, which typically require CT imaging. Therefore, the methodology presented in this study holds the potential for broader applicability, providing a valuable framework for studying recurrence across different cancer types.

The goal of this study was to preoperatively identify whether patients are likely to experience postoperative lung cancer recurrence. Specifically, predictive models were developed to estimate the probability of recurrence within 2 and 5 years after surgery. The AUC values for predicting recurrence within 2 years ranged from 0.676 to 0.737, while those for predicting recurrence within 5 years ranged from 0.738 to 0.770 (Figure 6). These results demonstrate the feasibility of using a sophisticated computer model to identify whether patients will or will not experience postoperative lung cancer recurrence.

The findings also suggest distinct factors may influence local/regional versus distant postoperative lung cancer recurrence. When the patients were grouped based on local/regional or distant postoperative recurrence, patients with local/regional recurrence had better RFS and OS at 5 years after surgery compared to patients with distant recurrence (Figure 5). The factors that were significantly associated with local/regional and distant recurrence were different (Table 2, Table 3, Table 4, Table 5, Table 6, Table 7, Table 8 and Table 9).

Various body tissues were significantly associated with RFS and OS. A comprehensive and objective analysis of body composition depicted on CT images revealed that SAT volume and density, SM density, and IMAT density are associated with RFS and OS. VAT density is only associated with OS. These data suggest that body composition may reveal insights into the mechanisms of postoperative lung cancer recurrence. Additionally, the pulmonary vasculature was also associated with RFS and OS, which is an interesting and unreported finding. These data suggest that body composition and pulmonary vasculature may reveal insights into the mechanisms of postoperative lung cancer recurrence, which warrants further investigation.

This study did not investigate what level of performance is necessary to be clinically meaningful, which was beyond the scope of this small retrospective investigation. The ROC analysis demonstrates that the factors analyzed in this study can be used to identify patients who will experience postoperative lung cancer recurrence (Figure 6). The balance is between improving the identification of patients who will experience recurrence (sensitivity) at the expense of misclassifying patients who will not experience recurrence (false positive). The decisions required to determine the clinical value and implementation of the computer model are a larger and deeper conversation.

Our findings suggest a higher risk of local or distant RFS in patients undergoing pneumonectomy compared to lobectomy (Table 2). The underlying reasons could be multifactorial. First, patients undergoing pneumonectomy often have more advanced disease or centrally located tumors, which inherently carry a higher tumor burden and metastatic potential. Additionally, the larger surgical trauma and loss of pulmonary reserve associated with pneumonectomy can impair postoperative recovery and immune response, creating a tumor-promoting environment. These patients may also experience greater difficulty tolerating adjuvant therapies, limiting their effectiveness. Finally, pneumonectomy may result in a higher release of circulating tumor cells (CTCs) and an increased likelihood of residual cancer cells, further contributing to recurrence risk.

This study has a few limitations that need to be addressed. The sample size was relatively small, and most of the patients were white. To mitigate the risk of overfitting, we employed the well-established k-fold cross-validation method. K-fold cross-validation offers several advantages, including efficient data utilization, as every data point is used for both training and validation, which is especially beneficial for small datasets. Additionally, it helps reduce overfitting by providing a more reliable performance estimate through averaging results across multiple folds, thus lowering the variance in performance metrics. Nevertheless, in practice, it is more desirable to evaluate the conclusions and the performance of the models using an external dataset. The use of different CT and PET scanners and protocols over a twelve-year period may introduce variability in the image data, which could have an impact on the results. Image factors that were significantly associated with postoperative lung cancer recurrence were identified despite the potential variance in image data quality. Surgical procedures could also change (improve) over a long time from the study, which was not believed to significantly impact the results. Survival was an outcome variable, which could have also been affected by something other than recurrence. In this study, death from all causes was used to determine overall survival because unreported or undetected recurrence may or may not have contributed to the death. Comorbidities (e.g., cardiovascular disease) and molecular tests were not evaluated in the factors related to recurrence. In a retrospective review of the medical records, it is not easy to identify potential comorbidities, and not all the patients have a complete mutation or molecular test. It is possible that undetected metastasis occurred prior to surgery, but it was detected after surgery and reported as a recurrence. In this study, there were no means to control for this confounding possibility. This study evaluated patients from a single institution, which may limit the generalizability of the findings to other patient populations from other institutions. The results of this study warrant further prospective and multisite investigation to confirm the findings.

Our future work will focus on validating the predictive models using larger, more diverse, and multi-institutional cohorts to enhance their generalizability and clinical applicability. Additionally, integrating molecular and genetic biomarkers with imaging data could provide deeper insights into the mechanisms driving postoperative recurrence. To further refine the models, we aim to standardize imaging protocols and leverage advanced AI techniques to improve their accuracy and robustness. Also, the primary objective of this study was to focus on preoperative imaging features, specifically from CT scans, to predict postoperative recurrence. We intentionally limited the scope to preoperative data to identify early biomarkers that could help guide clinical decision-making prior to surgery. However, we recognize that postoperative therapy plays a crucial role in patient outcomes and may influence the risk of recurrence. Therefore, we plan to include postoperative treatments, such as chemotherapy, immunotherapy, or radiation therapy, in our future work. Lastly, expanding these methodologies to other cancer types represents a promising direction for broader application and impact.

## 5. Conclusions

This study investigates the association between a wide range of factors and postoperative lung cancer recurrence and survival. A multi-level strategy was proposed to approach this clinically important problem with a special emphasis on body composition. Five different body composition tissues were automatically segmented and quantified to study their impact on lung cancer recurrence. The interesting findings are that detailed body composition and pulmonary vasculature are associated with recurrence-free survival and overall survival. The computer models of this study were able to successfully identify patients who did and did not experience postoperative lung cancer recurrence based on a wide range of factors, which indicates the complex mechanism and patient characteristics associated with postoperative recurrence.

## Figures and Tables

**Figure 1 cancers-17-00033-f001:**
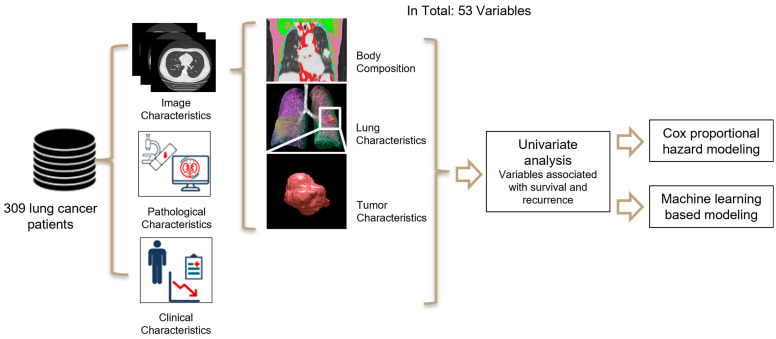
Multi-level radiomics strategy.

**Figure 2 cancers-17-00033-f002:**
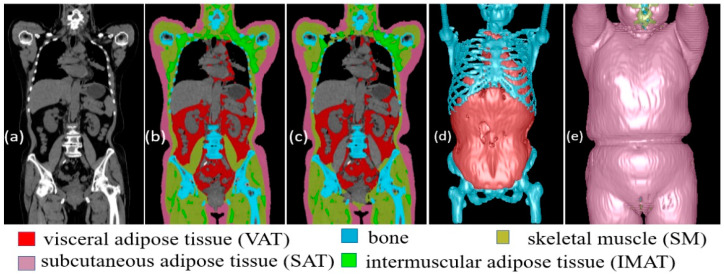
Automated segmentation of body tissue by CNN-based models and manual segmentation on a whole-body PET-CT scan. (**a**) The original CT image, (**b**) the manual annotations of the body tissues, and (**c**) the computer segmentations of the body tissues. (**d**,**e**) The 3D visualization of the five body tissues.

**Figure 3 cancers-17-00033-f003:**
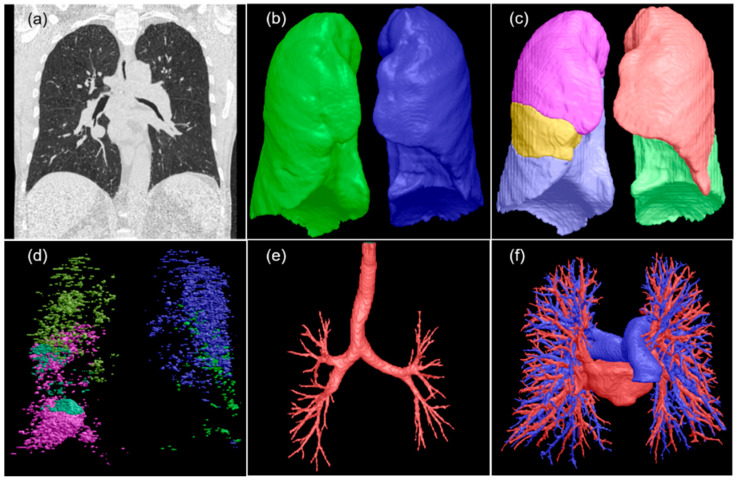
Segmentation of various lung structures. (**a**) The original CT image; (**b**–**f**) the 3D visualization of segmented lungs, lobes, emphysema densities, airways, and pulmonary arteries and veins.

**Figure 4 cancers-17-00033-f004:**
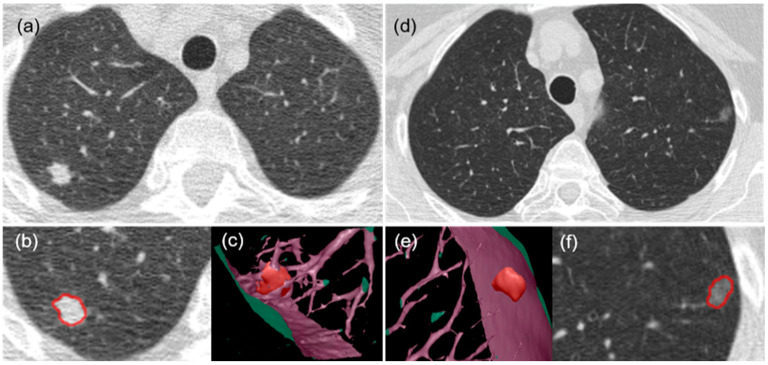
Segmentation of lung tumors on CT images. (**a**,**d**) The original CT images, (**b**,**f**) the contour of segmented tumors on the enlarged CT images, and (**c**,**e**) the 3D visualization of segmented tumors and surrounding areas.

**Figure 5 cancers-17-00033-f005:**
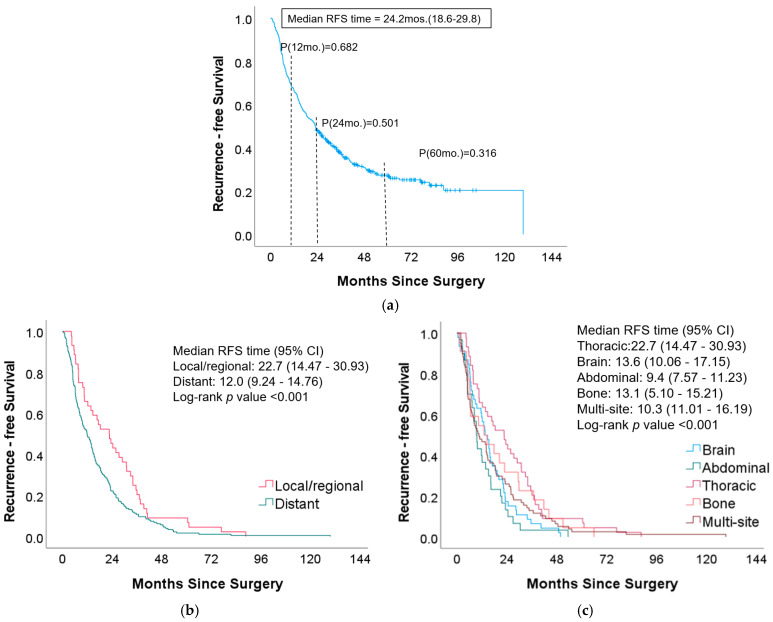
Kaplan–Meier curves for the recurrence-free survival (RFS) and the overall survival (OS) of the lung cancer patients after surgery: (**a**) overall RFS, (**b**,**c**) RFS grouped by regions and organs, respectively, and (**d**,**e**) OS grouped by regions and organs, respectively.

**Figure 6 cancers-17-00033-f006:**
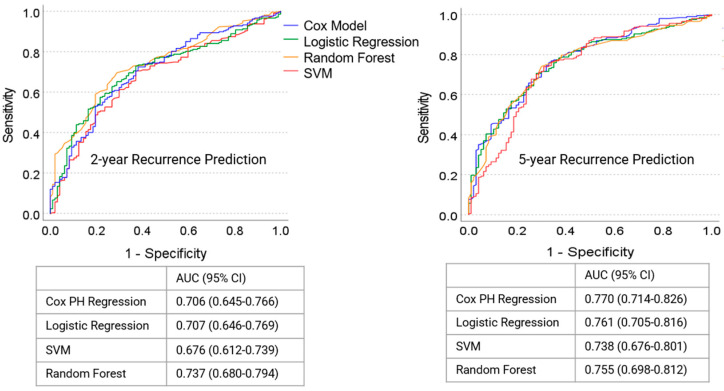
ROC curves of the computer models to identify patients who did or did not experience postoperative lung cancer recurrence within 2 and 5 years.

**Table 1 cancers-17-00033-t001:** Patient demographics (*n* = 309).

		No Recurrence (*n* = 90)		Local/Regional Recurrence (*n* = 44)	Distant Recurrence (*n* = 175)			
Characteristic		Alive (*n* = 10)	Expired (*n* = 80)		Brain (*n* = 46)	Abdominal (*n* = 30)	Bone (*n* = 22)	Multisite (*n* = 77)
Age, mean		70 ± 9.3	71 ± 9.8	65 ± 10.3	65 ± 10.3	68 ± 9.1	68 ± 8.6	67 ± 9.9
Height (cm)		167 ± 10.9	167 ± 9.9	170 ± 10.5	169 ± 11.0	167 ± 9.9	171 ± 10.9	169 ± 9.4
Weight (kg)		77 ± 20.4	77 ± 18.7	83 ± 20.4	79 ± 20.2	74 ± 15.4	83 ± 16.6	77 ± 19.2
BMI		27 ± 4.5	27 ± 6.11	29 ± 5.8	27 ± 6.1	26 ± 5.3	28 ± 4.9	27 ± 5.8
Sex								
	Female	7 (70.0)	47 (58.8)	22 (50.0)	24 (52.2)	15 (50)	11 (50.0)	38 (49.4)
	Male	3 (30.0)	33 (41.2)	22 (50.0)	22 (47.8)	15 (50)	11 (50.0)	39 (50.6)
Race								
	White	8 (80.0)	74 (92.5)	37 (84.1)	43 (93.5)	29 (96.7)	20 (90.9)	68 (88.3)
	Black	1 (10.0)	6 (7.5)	7 (15.9)	2 (4.4)	1 (3.3)	2 (9.1)	8 (10.4)
	Asian	1 (10.0)	0 (0.0)	0 (0.0)	1 (2.2)	0 (0.0)	0 (0.0)	1 (1.3)
Smoking status								
	Non-smoker	1 (10.0)	2 (2.5)	5 (11.4)	3 (6.5)	5 (16.7)	2 (9.1)	10 (13.0)
	Smoker	9 (90.0)	78 (97.5)	39 (88.6)	43 (93.5)	25 (83.3)	20 (90.9)	67 (87.0)
Surgical procedure								
	Lobectomy	10 (100.0)	63 (78.8)	28 (63.6)	36 (78.3)	23 (79.3)	17 (77.3)	62 (80.5)
	Segmentectomy	0 (0.0)	16 (20.0)	10 (22.7)	5 (10.9)	4 (13.8)	3 (13.6)	11 (14.3)
	Lobectomy & Segmentectomy	0 (0.0)	0 (0.0)	3 (6.8)	1 (2.2)	0 (0.0)	0 (0.0)	0 (0.0)
	Bilobectomy	0 (0.0)	1 (1.2)	3 (6.8)	2 (4.4)	0 (0.0)	0 (0.0)	1 (1.3)
	Pneumonectomy	0 (0.0)	0 (0.0)	0 (0.0)	2 (4.4)	3 (10)	2 (9.1)	3 (3.9)
Tumor location								
	RUL	4 (40.0)	39 (48.7)	17 (38.6)	21 (45.7)	13 (44.8)	11 (50.0)	25 (32.5)
	RML	0 (0.0)	5 (6.3)	4 (9.1)	5 (10.9)	1 (3.5)	2 (9.1)	3 (3.9)
	RLL	1 (10.0)	10 (12.5)	6 (13.6)	6 (13.0)	4 (13.8)	1 (4.6)	14 (18.2)
	LUL	2 (20.0)	16 (20.0)	9 (20.5)	8 (17.4)	6 (20.7)	5 (22.7)	24 (31.1)
	LLL	5 (50.0)	10 (12.5)	8 (18.2)	6 (13.0)	5 (17.2)	3 (13.6)	11 (14.3)
Lung cancer stage								
	0-	0 (0.0)	0 (0.0)	0 (0.0)	0 (0.0)	0 (0.0)	0 (0.0)	0 (0.0)
	IA-IB	9 (90.0)	58 (72.5)	23 (52.3)	12 (26.1)	8 (27.6)	11 (50.0)	37 (48.0)
	IIA-IIB	1 (10.0)	15 (18.7)	14 (31.8)	18 (39.1)	9 (31.0)	5 (22.7)	19 (24.7)
	IIIA-IIIB	0 (0.0)	7 (8.8)	7 (15.9)	16 (34.8)	12 (41.4)	6 (27.3)	21 (27.3)
Nodal involvement								
	N0	10 (100.0)	69 (85.2)	27 (61.4)	26 (56.5)	14 (46.7)	14 (63.6)	53 (68.8)
	N1	0 (0.0)	11 (14.8)	17 (38.6)	20 (43.5)	16 (53.3)	8 (36.4)	24 (31.2)
	N2	0 (0.0)	0 (0.0)	0 (0.0)	0 (0.0)	0 (0.0)	0 (0.0)	0 (0.0)

Data are numbers (%) unless otherwise noted. BMI: body mass index; RUL: right upper lobe; RML: right middle lobe; RLL: right lower lob; LUL: left upper lobe; LLL: left lower lobe.

**Table 2 cancers-17-00033-t002:** Univariate Cox PH analysis of recurrence-free survival (RFS) with demographic and clinical characteristics.

Variables	RFS		Local/Regional RFS		Distant RFS	
	*p* Value	Hazard Ratio * (95% CI)	*p* Value	Hazard Ratio * (95% CI)	*p* Value	Hazard Ratio * (95% CI)
Age	**<0.001**	**0.98 (0.97–0.99)**	0.435	0.99 (0.95–1.02)	0.114	0.99 (0.97–1.00)
Gender						
Female	0.051	1.31 (1.00–1.72)	0.737	0.89 (0.45–1.75)	0.722	0.91 (0.54–1.53)
Male	reference		reference		reference	
Race						
White	reference		reference		reference	
Others (Black/Asian)	0.688	1.10 (0.69–1.75)	0.794	1.15 (0.40–3.35)	0.722	0.91 (0.54–1.53)
Current/former smoker						
Yes	reference		reference		reference	
No	0.232	0.76 (0.49–1.19)	0.213	0.51 (0.18–1.47)	**0.021**	**0.57 (0.36–0.92)**
BMI	0.296	0.99 (0.96–1.01)	**0.022**	**1.06 (1.01–1.12)**	0.371	0.99 (0.96–1.01)
Tumor site						
RUL	reference		reference		reference	
RML	0.685	1.12 (0.65–1.94)	0.987	0.99 (0.22–4.49)	1.000	1.00 (0.55–1.81)
RLL	0.552	1.13 (0.99–1.98)	0.751	1.18 (0.43–3.21)	0.416	0.83 (0.53–1.31)
LUL	0.059	1.40 (0.99–1.98)	0.069	1.68 (1.11–6.48)	0.070	1.43 (0.97–2.09)
LLL	0.644	1.10 (0.74–1.64)	0.384	1.57 (0.57–4.30)	0.932	0.98 (0.63–1.53)
Surgical procedure						
Lobectomy	reference		reference		reference	
Segmentectomy	0.808	0.95 (0.65–1.39)	**<0.001**	**3.65 (1.70–7.84)**	0.180	0.73 (0.46–1.16)
Lobectomy & Segmentectomy	**0.036**	**2.91 (1.07–7.87)**	**<0.001**	**18.7 (5.12–68.5)**	0.421	0.45 (0.06–3.19)
Bisegmentectomy	0.438	1.35 (0.63–2.88)	0.993	0.99 (0.19–5.09)	0.200	0.55 (0.22–1.37)
Pneumonectomy	**<0.001**	**7.18 (3.57–14.4)**	**0.042**	**9.06 (1.08–75.8)**	**<0.001**	**4.23 (2.03–8.84)**

BMI: body mass index; RUL: right upper lobe; RML: right middle lobe; RLL: right lower lob; LUL: left upper lobe; LLL: left lower lobe. * Hazard ratio for lung cancer recurrence relative to the reference level (“reference”) for categorical variables. Variables with skewed distribution were log-transformed for the assessment. Bold text indicates variable is statistically significant.

**Table 3 cancers-17-00033-t003:** Univariate Cox PH analysis of recurrence-free survival (RFS) with pathological characteristics.

Variables	RFS		Local/Regional RFS		Distant RFS	
	*p* Value	Hazard Ratio * (95% CI)	*p* Value	Hazard Ratio * (95% CI)	*p* Value	Hazard Ratio * (95% CI)
Cancer stage						
I	**<0.001**	**0.29 (0.21–0.41)**	0.129	0.48 (0.19–1.24)	**<0.001**	**0.27 (0.19–0.38)**
II	**0.004**	**0.60 (0.42–0.85)**	0.749	0.85 (0.3–2.35)	**0.004**	**0.57 (0.39–0.83)**
III	reference		reference		reference	
Nodal involvement						
N0	**<0.001**	**0.44 (0.33–0.58)**	**0.002**	**0.34 (0.17–0.67)**	**<0.001**	**0.46 (0.34–0.63)**
N1/N2	reference		reference		reference	
Histological subtype						
Adenocarcinoma	**0.001**	**0.58 (0.42–0.81)**	**0.044**	**0.39 (0.15–0.98)**	**<0.001**	**0.42 (0.29–0.61)**
Squamous cell carcinoma	**<0.001**	**0.38 (0.24–0.59)**	0.837	1.12 (0.37–3.40)	0.078	0.63 (0.38–1.05)
Others	reference		reference		reference	

Cancer stage: overall pathological stage. * Hazard ratio for lung cancer recurrence relative to the reference level (“reference”). Variables with skewed distribution were log-transformed for the assessment. Bold text indicates variable is statistically significant.

**Table 4 cancers-17-00033-t004:** Univariate Cox PH analysis of recurrence-free survival (RFS) with body composition characteristics.

Variables	RFS		Local/Regional RFS		Distant RFS	
	*p* Value	Hazard Ratio * (95% CI)	*p* Value	Hazard Ratio * (95% CI)	*p* Value	Hazard Ratio * (95% CI)
VAT volume (L)	0.356	0.95 (0.84–1.07)	0.209	1.17 (0.92–1.48)	0.118	0.90 (0.78–1.03)
VAT density (HU)	0.146	1.01 (1.00–1.03)	0.450	0.98 (0.93–1.03)	0.056	1.02 (1.00–1.04)
SAT volume (L)	0.117	0.96 (0.90–1.01)	**0.012**	**1.14 (1.03–1.25)**	**0.004**	**0.91 (0.85–0.97)**
SAT density (HU)	**0.017**	**1.11 (1.03–1.22)**	0.409	0.99 (0.95–1.02)	**0.003**	**1.12 (1.01–1.13)**
IMAT volume (L)	0.320	0.77 (0.47–1.28)	0.084	2.54 (0.88–7.31)	0.102	2.68 (1.00–3.10)
IMAT density (HU)	**0.022**	**1.42 (1.11–1.64)**	0.968	1.00 (0.97–1.04)	**0.013**	**1.23 (1.01–1.35)**
SM volume (L)	0.125	1.07 (0.98–1.16)	0.192	1.14 (0.94–1.39)	0.271	1.05 (0.96–1.16)
SM density (HU)	**0.017**	**1.33 (1.31–1.34)**	0.782	1.01 (0.97–1.04)	**<0.001**	**1.03 (1.02–1.05)**
Bone volume (L)	0.189	1.19 (0.92–1.53)	0.345	1.33 (0.74–2.40)	0.310	1.16 (0.87–1.53)
Bone density (HU)	0.136	1.00 (1.00–1.00)	0.870	1.00 (0.99–1.01)	0.120	1.00 (1.00–1.01)

Visceral adipose tissue (VAT), subcutaneous adipose tissue (SAT), intermuscular adipose tissue (IMAT), skeletal muscle (SM). * Hazard ratio for lung cancer recurrence relative to the reference level (“reference”). Variables with skewed distribution were log-transformed for the assessment. Bold text indicates variable is statistically significant.

**Table 5 cancers-17-00033-t005:** Univariate Cox PH analysis of recurrence-free survival (RFS) with CT-computed lung and tumor characteristics.

Variables	RFS		Local/regional RFS		Distant RFS	
	*p* Value	Hazard Ratio * (95% CI)	*p* Value	Hazard Ratio * (95% CI)	*p* Value	Hazard Ratio * (95% CI)
Tumor volume (ML)	**<0.001**	**1.17 (1.11–1.24)**	0.499	0.92 (0.72–1.18)	**<0.001**	**1.19 (1.13–1.26)**
Tumor ground glass opacity	**0.002**	**0.04 (0.01–0.86)**	0.669	0.26 (0.00–1.21)	**0.002**	**0.003 (0–0.111)**
Tumor irregularity	**0.006**	**2.29 (1.27–4.12)**	0.902	0.02 (0.00–0.90)	**0.002**	**2.73 (1.43–5.19)**
Pleural area (cm^2^)	**<0.001**	**1.04 (1.02–1.06)**	0.726	1.01 (0.95–1.08)	**<0.001**	**1.04 (1.03–1.06)**
Pulmonary artery vol. (L)	**0.003**	**1.41 (1.14–1.76)**	0.688	0.15 (0.00–1.72)	**0.001**	**1.42 (1.27–1.85)**
Pulmonary vein vol. (L)	**0.003**	**2.46 (1.53–3.42)**	0.817	0.35 (0.00–1.34)	**0.002**	**1.16 (1.04–1.58)**

* Hazard ratio for lung cancer recurrence relative to the reference level (“reference”). Variables with skewed distribution were log-transformed for the assessment. Bold text indicates variable is statistically significant.

**Table 6 cancers-17-00033-t006:** Univariate Cox PH analysis of overall survival (OS) with demographic and clinical characteristics.

Variables	OS		Local/Regional OS		Distant OS	
	*p* Value	Hazard Ratio * (95% CI)	*p* Value	Hazard Ratio * (95% CI)	*p* Value	Hazard Ratio * (95% CI)
Age	0.203	0.99 (0.98–1.01)	0.279	1.02 (0.98–1.06)	0.754	1.00 (0.99–1.02)
Gender						
Female	0.373	0.89 (0.71–1.11)	0.685	1.15 (0.58–2.31)	0.093	1.29 (0.96–1.72)
Male	reference		reference		reference	
Race						
White	reference		reference		reference	
Others (Black and Asian)	0.682	0.91 (0.57–1.44)	0.872	1.09 (0.38–3.12)	0.530	0.85 (0.51–1.42)
Current/former smoker						
Yes	reference		reference		reference	
No	0.284	1.26 (0.89–1.78)	0.217	1.98 (0.67–5.88)	0.133	1.44 (0.89–2.33)
BMI	0.085	0.99 (095–1.01)	0.082	1.06 (0.99–1.12)	**0.018**	**0.97 (0.94–0.99)**
Tumor site						
RUL	reference		reference		reference	
RML	0.861	1.05 (0.61–1.82)	0.397	1.96 (0.41–9.30)	0.350	1.33 (0.73–2.41)
RLL	0.613	1.11 (0.74–1.67)	0.321	1.68 (0.61–4.64)	0.637	0.90 (0.56–1.42)
LUL	0.522	1.12 (0.79–1.58)	0.076	1.53 (1.44–8.68)	0.056	1.45 (0.99–2.13)
LLL	0.937	0.98 (0.66–1.48)	0.197	1.95 (0.71–5.37)	0.969	1.01 (0.65–1.57)
Surgical Procedure						
Lobectomy	reference		reference		reference	
Segmentectomy	0.725	0.93 (0.64–1.37)	0.060	2.75 (1.28–5.92)	0.155	0.72 (0.45–1.13)
Lobectomy & Segmentectomy	0.935	0.96 (0.35–2.68)	0.049	3.54 (1.01–12.4)	0.113	0.20 (0.03–1.46)
Bisegmentectomy	0.594	1.23 (0.58–2.62)	0.290	2.21 (051–9.60)	0.473	0.72 (0.30–1.76)
Pneumonectomy	**0.012**	**2.38 (1.21–4.66)**	0.565	1.81 (0.24–13.8)	**0.024**	**2.28 (1.11–4.66)**

BMI: body mass index; RUL: right upper lobe; RML: right middle lobe; RLL: right lower lob; LUL: left upper lobe; LLL: left lower lobe. * Hazard ratio for lung cancer recurrence relative to the reference level (“reference”) for categorical variables. Variables with skewed distribution were log-transformed for the assessment. Bold text indicates variable is statistically significant.

**Table 7 cancers-17-00033-t007:** Univariate Cox PH analysis of overall survival (OS) with pathological characteristics.

Variables	OS		Local/Regional OS		Distant OS	
	*p* Value	Hazard Ratio * (95% CI)	*p* Value	Hazard Ratio * (95% CI)	*p* Value	Hazard Ratio * (95% CI)
Cancer stage						
I	**<0.001**	**0.47 (0.34–0.65)**	0.286	1.66 (0.66–4.21)	**<0.001**	**0.41 (0.29–0.58)**
II	0.277	0.82 (0.58–1.17)	0.365	1.60 (0.58–4.40)	0.167	0.77 (0.53–1.12)
III	reference		reference		reference	
Nodal involvement						
N0	**<0.001**	**0.58 (0.44–0.76)**	0.698	0.87 (0.44–1.74)	**<0.001**	**0.58 (0.43–0.78)**
N1/N2	reference		reference		reference	
Histological subtype						
Adenocarcinoma	0.597	0.92 (0.66–1.27)	0.664	0.82 (0.33–2.03)	**0.023**	**0.66 (0.46–0.94)**
Squamous cell carcinoma	**0.041**	**0.62 (0.39–0.98)**	0.497	1.16 (0.49–4.37)	0.332	0.78 (0.47–1.29)
Others	reference		reference		reference	

Cancer stage: overall pathological stage. * Hazard ratio for lung cancer recurrence relative to the reference level (“reference”). Variables with skewed distribution were log-transformed for the assessment. Bold text indicates variable is statistically significant.

**Table 8 cancers-17-00033-t008:** Univariate Cox PH analysis of overall survival (OS) with body composition.

Variables	OS		Local/Regional OS		Distant OS	
	*p* Value	Hazard Ratio * (95% CI)	*p* Value	Hazard Ratio * (95% CI)	*p* Value	Hazard Ratio * (95% CI)
VAT volume (L)	0.278	0.94 (0.84–1.05)	0.367	1.09 (0.90–1.31)	0.138	0.90 (0.79–1.03)
VAT density (HU)	**0.033**	**1.02 (1.00–1.04)**	0.438	0.98 (0.93–1.03)	0.091	1.02 (1.00–1.04)
SAT volume (L)	**0.018**	**0.93 (0.88–0.99)**	0.095	1.07 (0.99–1.17)	**<0.001**	**0.88 (0.82–0.95)**
SAT density (HU)	**<0.001**	**1.02 (1.01–1.03)**	0.845	0.97 (0.96–1.03)	**<0.001**	**1.02 (1.01–1.04)**
IMAT volume (L)	0.230	0.74 (0.45–1.21)	0.188	1.73 (0.77–3.91)	0.143	0.67 (0.40–1.14)
IMAT density (HU)	**0.002**	**1.03 (1.01–1.05)**	0.637	1.01 (0.96–1.07)	**0.002**	**1.03 (1.01–1.06)**
SM volume (L)	0.397	1.04 (0.96–1.12)	0.842	1.01 (0.85–1.22)	0.322	0.96 (0.87–1.05)
SM density (HU)	**0.002**	**1.02 (1.01–1.03)**	0.310	0.98 (0.95–1.02)	**0.001**	**1.02 (1.01–1.04)**
Bone volume (L)	0.176	1.18 (0.93–1.51)	0.746	1.10 (0.62–1.94)	0.626	0.94 (0.72–1.22)
Bone density (HU)	0.348	1.00 (1.00–1.00)	0.483	1.00 (0.99–1.01)	0.747	1.00 (1.00–1.00)

Visceral adipose tissue (VAT), subcutaneous adipose tissue (SAT), intermuscular adipose tissue (IMAT), skeletal muscle (SM). * Hazard ratio for lung cancer recurrence relative to the reference level (“reference”). Variables with skewed distribution were log-transformed for the assessment. Bold text indicates variable is statistically significant.

**Table 9 cancers-17-00033-t009:** Univariate Cox PH analysis of overall survival (OS) with CT-computed lung and tumor characteristics.

Variables	OS		Local/Regional OS		Distant OS	
	*p* Value	Hazard Ratio * (95% CI)	*p* Value	Hazard Ratio * (95% CI)	*p* Value	Hazard Ratio * (95% CI)
Tumor volume (ML)	**0.023**	**1.09 (1.04–1.16)**	0.086	0.99 (0.97–1.00)	**<0.001**	**1.12 (1.06–1.19)**
Tumor ground glass opacity	**0.039**	**0.04 (0.00–0.86)**	0.347	1.52 (0.10–2.74)	**0.018**	**0.02 (0.00–0.48)**
Tumor irregularity	**0.029**	**2.01 (1.07–3.63)**	0.491	0.55 (0.10–3.01)	**0.010**	**2.38 (1.23–4.62)**
Pleural area	0.211	1.01 (0.99–1.03)	0.170	0.96 (0.90–1.02)	**0.044**	**1.02 (1.00–1.03)**
Pulmonary artery vol. (L)	**0.009**	**1.83 (1.41–2.34)**	0.197	0.01 (0.00–1.11)	**0.001**	**2.42 (1.67–2.96)**
Pulmonary vein vol. (L)	**0.040**	**1.62 (1.21–1.82)**	0.301	0.01 (0.00–1.17)	**0.016**	**1.97 (1.72–2.16)**

* Hazard ratio for lung cancer recurrence relative to the reference level (“reference”). Variables with skewed distribution were log-transformed for the assessment. Bold text indicates variable is statistically significant.

**Table 10 cancers-17-00033-t010:** Multivariate Cox PH analysis of recurrence-free survival (RFS).

Variables		RFS		Local/Regional RFS		Distant RFS	
		*p* Value	Hazard Ratio (95% CI)	*p* Value	Hazard Ratio (95% CI)	*p* Value	Hazard Ratio (95% CI)
Surgical procedure							
	Lobectomy	reference		reference		reference	
	Pneumonectomy	<0.001	4.20 (1.96–8.99)	<0.001	6.09 (2.90–12.80)	<0.001	3.93 (1.84–8.38)
Cancer stage							
	I	<0.001	0.33 (0.23–0.48)	-	-	<0.001	0.32 (0.22–0.48)
	II	0.013	0.62 (0.43–0.91)	-	-	0.002	0.54 (0.36–0.80)
	III	reference		reference		reference	
Nodal involvement							
	N0	<0.001	0.51 (0.38–0.69)	-	-	-	-
	N1/N2	reference		reference		reference	
SM density (HU)		0.002	1.02 (1.01–1.04)	-	-	0.020	1.02 (1.00–1.04)
Pulmonary artery vol. (L)						0.004	2.69 (1.16–3.36)
Pulmonary vein vol. (L)		0.025	1.12 (1.06–1.34)	-	-	-	-

“-”: *p* > 0.05.

**Table 11 cancers-17-00033-t011:** Multivariate Cox PH analysis of overall survival (OS).

Variables		OS		Local/Regional OS		Distant OS	
		*p* value	Hazard Ratio (95% CI)	*p* value	Hazard Ratio (95% CI)	*p* value	Hazard Ratio (95% CI)
Cancer stage							
	I	-	-	-	-	<0.001	0.49 (0.34–0.71)
	II	-	-	-	-	-	-
	III	reference		reference		reference	
Nodal involvement							
	N0	<0.001	0.58 (0.44–0.78)				
	N1/N2	reference		reference		reference	
SAT density (HU)		<0.001	1.02 (1.01–1.04)	-	-	<0.001	1.02 (1.01-1.04)
Pulmonary artery vol. (L)		0.028	2.32 (1.52–2.70)	-	-	0.011	1.23 (1.01–1.36)

“-”: *p* > 0.05.

## Data Availability

The datasets presented in this article are not readily available because the data are part of an ongoing study.

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
