# Peer review of "Predicting Postoperative Lung Cancer Recurrence and Survival Using Cox Proportional Hazards Regression and Machine Learning"

_cancers, 2024, doi:10.3390/cancers17010033_

Round 1
Reviewer 1 Report
Comments and Suggestions for Authors
This study constructs a predictive model for predicting NSCLC recurrence by analyzing radiological features derived from CT scans. Overall, the manuscript is well organized. Here are some questions:
1. It is interesting that patients underwent pneumonectomy had a higher risk of local or distant RFS than lobectomy. Is this relevant to postoperative recovery or physical status? Or patients receiving pneumonectomy had an advanced stage of cancer, which causes poor survival.
2. Subtitle of results section is recommended. In the current version of manuscript, authors only use narrative words to describe the results of tables and figures, while indication or conclusive statements are not presented.
3. When comparing the effect of histological subtype, what is others? This information is not provided in Table 1.
4. It seems that most parameters are significantly associated with the prognosis of patients with distant recurrence rather than local recurrence. From current finding, which feature matters most for predicting local/regional recurrence?
Minor:
1. The description of “In the study cohort, 70.9% (219/309) of the subjects with NSCLC experienced postoperative cancer recurrence, 20.1% (44/219) of the subjects experienced local/regional cancer recurrence, 79.9% (175/219) of the subjects experienced distant cancer recurrence, and 88.9% (80/90) of the subjects expired without evidence of recurrence (Table 1)” is ambiguous. Those recurrence should be “among which” or “where”…. before “20.1%”.
2. Proofreading is necessary. Some abbreviations are defined twice and grammar issues are detected.
3. The definition of RFS should be provided in methods rather than results. And methodology should be put in methods, eg. “A p-value of less than 0.05 (bold and underlined) is considered statistically significant, indicating that variables are related to longer OS if the hazard ratio is less than 1. Conversely, if the hazard ratio is greater than 1, the variables are associated with shorter OS.”
Reviewer 2 Report
Comments and Suggestions for Authors
The paper evaluates a study that predicts lung cancer recurrence and survival post-surgery using Cox regression and machine learning. It analyzes preoperative chest CT scans of 309 NSCLC patients, focusing on image biomarkers and factors such as surgical details and tumor traits. Both models demonstrated similar accuracy, with an AUC of 0.75-0.77. The study calls for further validation with a larger, more diverse cohort to enhance reliability and acknowledges existing limitations:
1. Your article does not specify the software environment used. Could you please provide full details about it?
2. Please write more information about use of "Dataset.".
3. What evaluation metrics do you use in your work?
4. My only concern is that the Discussion section is completely inadequate.
5. Please add more information about future work.
6. The method section introduces various models used in the study and provides detailed information on how to use them.
7. It is a good idea to compare the results with other models, but it is necessary to summarize only the results for the developed model.
Reviewer 3 Report
Comments and Suggestions for Authors
This article aims to identify image biomarkers from preoperative chest CT scans to predict recurrence in patients who had lung resection by comparing Cox proportional hazard regression analysis with machine learning (ML) methods for predicting recurrence.
The paper is written clearly and concisely. The introduction clearly outlines the scope and objectives, the aims and research questions of the review paper clearly stated. The references are accurate and complete. Overall, this work is suitable for publication after addressing a few minor concerns.
Comments and suggestions:
1. All the 309 patients’ samples who underwent lung resection were selected as the study cohort. Using part of the patients’ samples to validate this method is also important, although the author cannot further validate through external validation using a larger cohort from multiple sites.
2. After surgery, whether the patients receive therapy should be considered as a factor to predict recurrence.
3. In Figure 1, change the brain image figure in the ‘Image Characteristics’ to a lung cancer image.
4. Move Figure 4a and 4d upper to Figure 4b-c, Figure 4e-f.
5. The title of ‘2.6. Performance validation’ should not be bold.
Overall, I recommend minor revisions to this review before its publication in Cancers.
Round 2
Reviewer 1 Report
Comments and Suggestions for Authors
Authors address my questions.